# Effects of fasting on isolated murine skeletal muscle contractile function during acute hypoxia

Cameron A. Schmidt[1,2], Emma J. Goldberg[1,2], Tom D. Green[1,2], Reema R. Karnekar[1,2], Jeffrey J. Brault[1,3], Spencer G. Miller[3], Adam J. Amorese[1,2], Dean J. Yamaguchi[4,5], Espen E. Spangenburg[1,2], Joseph M. McClung[1,2,4]*

1 Dept. of Physiology, Brody School of Medicine, East Carolina University, Greenville, North Carolina, United States of America, 2 East Carolina Diabetes and Obesity Institute, East Carolina University, Greenville, North Carolina, United States of America, 3 Dept. of Anatomy and Cell Biology, Indiana University School of Medicine, Indianapolis, Indiana, United States of America, 4 Department of Cardiovascular Sciences, East Carolina University, Greenville, North Carolina, United States of America, 5 Division of Surgery, Brody School of Medicine, East Carolina University, Greenville, North Carolina, United States of America

* mcclungj@ecu.edu

**Data Availability Statement:** All relevant data are available at DOI 10.17605/OSF.IO/PZT5G.

**Funding:** This work was made possible by grants awarded by the National Institutes of Health (NIH.

## Abstract

Stored muscle carbohydrate supply and energetic efficiency constrain muscle functional capacity during exercise and are influenced by common physiological variables (e.g. age, diet, and physical activity level). Whether these constraints affect overall functional capacity or the timing of muscle energetic failure during acute hypoxia is not known. We interrogated skeletal muscle contractile properties in two anatomically distinct rodent hindlimb muscles that have well characterized differences in energetic efficiency (locomotory- extensor digitorum longus (EDL) and postural- soleus muscles) following a 24 hour fasting period that resulted in substantially reduced muscle carbohydrate supply. 180 mins of acute hypoxia resulted in complete energetic failure in all muscles tested, indicated by: loss of force production, substantial reductions in total adenosine nucleotide pool intermediates, and increased adenosine nucleotide degradation product—inosine monophosphate (IMP). These changes occurred in the absence of apparent myofiber structural damage assessed histologically by both transverse section and whole mount. Fasting and the associated reduction of the available intracellular carbohydrate pool (~50% decrease in skeletal muscle) did not significantly alter the timing to muscle functional impairment or affect the overall force/work capacities of either muscle type. Fasting resulted in greater passive tension development in both muscle types, which may have implications for the design of pre-clinical studies involving optimal timing of reperfusion or administration of precision therapeutics.

## Introduction

Ischemic skeletal muscle necrosis occurs concurrently with several common clinical conditions (e.g. peripheral arterial disease, compartment syndrome, or diabetic necrosis) and is a complicating factor of successful muscle graft transplantation [1–3]. The severity of necrosis during an ischemic episode has long been considered a sole function of time, temperature, and

gov) to JMM (R01HL125695), EES
(R01AR066660) and JJB (R01AR070200).

**Competing interests:** The authors have declared
that no competing interests exist.

magnitude of the hypoxic insult [4,5]. However, the timing of the events that precede irreversible functional impairment and necrosis during ischemia may also depend on other key variables including: metabolic rate; contractile efficiency; and the size of the stored carbohydrate pool [4]. Carbohydrate metabolism is key, as muscle energy supply becomes dependent on anaerobic fermentation of stored carbohydrate sources during ischemia [6–8]. Glycogen is the primary storage form of carbohydrate in skeletal muscle, and its storage/utilization can be influenced by acute environmental factors as well as chronic diseases [9–14].

Previous studies have examined the time dependent changes of metabolites and contractile function in rodent skeletal muscle following ischemia with reperfusion (I/R) [15–18]. Several important observations can be gleaned from these studies: First, locomotory (fast glycolytic) muscles experienced more damage compared to postural (slow oxidative) muscles [15,17]. Second, The degree of initial injury can have large effects on post ischemic recovery time [16]. Lastly, optimal reperfusion timing is related to changes in muscle metabolite levels during ischemia [18]. A major limitation of I/R studies is that it is difficult to distinguish between the functional impairment and/or damage that is attributable to the ischemia itself versus that caused by the reperfusion injury.

In a previous study, using an *in vivo* mouse hindlimb ischemia model (without reperfusion), we found that myonecrosis develops between three and six hours after the onset of ischemia and is accompanied by a complete loss of contractile function [19]. This led us to examine the <3-hour time domain in this study to better define the timing of muscle functional impairments and associated terminal metabolite changes that occur during acute hypoxia. We hypothesized that fasting, and associated reductions in stored muscle glycogen, would significantly shorten the amount of time that the muscles could remain functional during acute hypoxia.

To test this hypothesis, we utilized fasting to induce an approximate 50% decrease in resting muscle glycogen and employed a carefully controlled experimental system to assess the effects of carbohydrate reduction on isolated mouse hindlimb muscle function during acute hypoxia. Our data provide a novel characterization of hypoxic muscle mechanical/energetic failure and paint a detailed picture of the timing of these impairments. This information can be used in conjunction with existing *in vivo* rodent hindlimb ischemia/reperfusion studies [5,16,17,19] to generate new hypotheses regarding optimal timing of reperfusion or administration of precision therapeutics.

## Materials and methods

### Animals

Adult male BALB/c mice (N = 32), aged 16–24 weeks old, were obtained from Jackson Laboratories (Bar Harbor, ME). All work was approved by the Institutional Animal Care and Use Committee of East Carolina University. Animal care followed the Guide for the Care and Use of Laboratory Animals, Institute of Laboratory Animal Resources, Commission on Life Sciences, National Research Council. Washington: National Academy Press, 1996. Animals had free access to water and food except during fasting protocols, during which animals had free access to water only.

### Laser scanning confocal microscopy

Sarcomeric actin staining was performed in PFA fixed whole mount muscles, following permabilization with 30μg/ml saponin, using 200nM Alexa Fluor 488 conjugated phalloidin (Thermo Fisher, Waltham MA). Muscles were imaged in a glass bottom (#1.5) dish in Krebs Ringer solution. All imaging was performed using an Olympus FV1000 laser scanning

confocal microscope (LSCM). Acquisition software was Olympus FluoView FSW (V4.2). The objective used was 60X oil immersion (NA = 1.35, Olympus Plan Apochromat UPLSAPO60X (F)) or 30X (NA = 1.05, Olympus Plan Apochromat UPLSAPO30XS). Images were 800x800 pixel with 2μs/pixel dwell time. Detector noise was reduced by application of a 3X line scanning kalman filter. Images were acquired in sequential scan mode. 2μM DAPI was used for nuclear counterstaining (Sigma Aldrich, St. Louis, MO) and was excited using the 405nm line of a multiline argon laser; emission was filtered using a 490nm dichroic mirror and 430-470nm barrier filter. AF488-phalloidin was excited using the 488nm line of a multiline argon laser; emission was filtered using a 560nm dichroic mirror and 505-540nm barrier filter. Zero detector offset was used for all images. The pinhole aperture diameter was set to 105um (1 Airy disc).

## Dystrophin/laminin immunofluorescence in transverse muscle sections

EDL and soleus muscles were embedded in optimal cutting temperature medium (OCT), and frozen in liquid nitrogen cooled isopentane for cryosectioning. 10μm sections were cut using a CM-3050S cryostat (Leica, Wetzlar Germany) and collected on charged glass slides. Sections were then fixed in 1:1 acetone/methanol for 10 minutes at -20˚C, rehydrated in 1X phosphate buffered saline (PBS), and blocked in 5% goat serum + 1X PBS for one hour at room temperature. Sections were then incubated with mouse anti-human monoclonal dystrophin antibody (Thermo-Fisher, MA5-13526), and rabbit anti-rat primary laminin antibody (Thermo-Fisher, A5-16287) at 4˚C overnight. Sections were washed 3X for 10 minutes with cold 1X PBS and incubated for 1 hour with Alexa-fluor 594 conjugated goat anti-rabbit IgG or Alexa-fluor 488 conjugated goat anti-mouse (highly cross adsorbed) IgG2b secondary antibody (1:250, Invitrogen). Sections were mounted using Vectashield hard mount medium without Dapi (Vector Labs). Images were taken with an Evos FL auto microscope (Thermo Fisher, Waltham, MA) with a plan fluorite 20X cover slip corrected objective lens (NA = 0.5, air). The following excitation/emission filter cubes were used: GFP (470/22 nm Excitation; 510/42 nm Emission) and Texas Red (585/29 nm Excitation; 624/40 nm Emission). 4X and 20X magnification images were taken for each condition. Image processing was performed using ImageJ (NIH, v1.51f) [20].

## Fasting

Mice were housed in a temperature-controlled facility on a 12-hour light-dark cycle with free access to food and water prior to fasting (dark cycle: beginning at 1900 hours, ending at 0700 hours). Mice were fasted for 24 hours to achieve a reduction of skeletal muscle glycogen of ~50%, compared to the fed state. The 24-hour fasting period began at the beginning of a light cycle (0700 hours) and was terminated at the end of the subsequent dark cycle (0700 hours). Mice had free access to water during fasting. All muscles (including control and fasted groups) were isolated for experiments immediately following the end of the dark cycle, between 0700 and 0800 hours. All experiments were performed in the summer season, between the months of May and August.

## Measurement of muscle mechanical function

Mice were sacrificed by cervical dislocation under isoflurane anesthesia (confirmed by lack of pedal withdrawal reflex). Extensor digitorum longus (EDL) or soleus muscles were carefully dissected and tied at both tendon ends with 5–0 silk sutures (Thermo Fisher, Waltham, MA). Muscles were tied to an anchor at the proximal end and a dual mode force transducer (Aurora 300B-LR, Aurora, ON, Canada) at the distal end in a vertical bath at 22˚C. All protocols were performed in the absence of additional carbon fuel sources (i.e. amino acids, glucose, etc.) to restrict muscles to stored fuel supplies. The bath solution was a modified Krebs Ringer solution

described previously [21]. All muscles were dissected and mounted within 15 mins of sacrifice. Muscles were equilibrated in the bath for 10 mins, and optimal length ($L_0$) was determined by stimulating twitch contractions (0.2ms pulse width, 1 pulse/train) at 10 second intervals and adjusting the length incrementally until maximal force was achieved. Supramaximal stimulation voltage for both muscle types was determined to be 20V. $L_O$ (mm) was measured using a digital microcaliper (Thermo Fisher, Waltham, MA). A force frequency curve was developed for each muscle using stepwise increasing stimulation frequencies of 10, 20, 40, 60, 80, 100, and 120 Hz (.2ms pulse width, pulses/train = half of the stim. Freq.). Baths were aerated with 95%$O_2$/5%$CO_2$ (oxygenated; $O_2$ condition) during $L_O$ determination and the initial force frequency curve. The aeration source was then either left the same or changed to 95%$N_2$/5%$CO_2$ (hypoxic; $N_2$ condition) to simulate an ischemia-like condition. The muscles were then equilibrated for 10 mins, and an initial isokinetic contraction protocol was elicited in the $O_2$ condition (100Hz isometric contraction for 0.8 seconds followed by a 3mm shortening phase over .3 seconds, then a return to $L_o$ over 30s for the EDL; for the soleus 80Hz isometric contraction was elicited for 0.8 seconds followed by a 4mm shortening phase over .4 seconds, then a return to $L_o$ over 30s). The aeration source was then either left the same or changed to 95%$N_2$/5% $CO_2$ (hypoxic condition), with experimental conditions alternated each time to reduce bias. The muscles were then equilibrated for 10 mins, followed by stimulated isokinetic contractions every 10 mins for 180 mins (18 total contractions). We chose this timing based on our previous observation that excitation contraction coupling is impaired in muscles isolated from BALB/c mice 180 minutes after *in vivo* induction of acute hindlimb ischemia (in the absence of histological signs of tissue necrosis) [19]. A second force frequency curve was measured following the 180-min. isokinetic protocol without changing the aeration source. Muscles were removed from the apparatus, blot dried on paper, weighed, and flash frozen in liquid nitrogen for biochemical analyses. Isometric time-tension integrals (TTI) were calculated by integrating over the isometric (phase I) portion of the curve and are expressed in units of Newton*second/ square centimeter ($N^*s/cm^2$). Isokinetic work (W) was obtained by integrating the force over the length change during the shortening (phase II) portion of the protocol and is expressed in units of Joules/square centimeter ($J/cm^2$).

Absolute isometric force measurements were normalized to mathematically approximated cross-sectional areas of the muscles. The cross-sectional area for each muscle was determined by dividing the mass of the muscle in grams (g) by the product of its optimal fiber length ($L_f$, cm) and estimated muscle density (1.06 g $cm^{-3}$). Muscle force production was expressed as specific force ($N/cm^2$) determined by dividing the tension in Newtons (N) by the calculated muscle cross-sectional area. $L_f$ was obtained by multiplying $L_O$ by the standard muscle length to fiber length ratio (0.45 for adult mouse EDL; 0.71 for soleus) [22]. A gas calibrated Clark electrode (Innovative instruments, Lake Park, NC) was used to assess the oxygen saturation of the isolated bath medium under both aeration conditions prior to carrying out the experiments. $O_2$ conditions were approximately 90% saturation measured at the center of the bath (after 10 mins of aeration). $N_2$ conditions were <2% saturation.

## Measurement of glycogen content in whole tissue

Skeletal muscle and liver tissues were flash frozen in liquid nitrogen and stored at -80˚C. Glycogen assays were performed using acid hydrolysis and an enzyme coupled assay [23]. Briefly, tissue samples were digested/hydrolyzed under acidic conditions using 2N hydrochloric acid (Sigma Aldrich, St. Louis, MO) on a heating block at 95˚C for 2 hours with additional vortexing. Samples were neutralized with equal volume 2N sodium hydroxide (Sigma). A small amount of tris HCl pH 7.0 (~1% of final volume) was added to buffer the solution. Samples

were added to a clear 96 well plate in duplicate and were incubated with a solution containing: >2000U/L hexokinase (S. *cerevisiae*), >4000 U/L $NAD^+$ dependent glucose-6-phosphate dehydrogenase (L. *mesenteroides*), 4mM ATP, 2mM $Mg^{2+}$, and 2mM $NAD^+$ (Hexokinase reagent solution; Thermo Fisher). Water was used in place of the reagent for background correction. A standard curve of D-glucose (Sigma Aldrich) was used to calculate the concentrations of hydrolyzed glucosyl units in each sample. Colorimetric measurement of NAD(P)H absorbance was made at 340nm using a Cytation 5 microtiter plate reader (Biotek, Winooski, VT). Liver samples were diluted 1:50 in water prior to enzyme coupled assays to obtain absorbance values within the range of the standard curve. Data were normalized to tissue mass and represented as nmoles glycogen/mg tissue wet weight. The response coefficient ($R_{Glyc}$) is defined as the fractional change in experimental group mean relative to the basal group (i.e. Mean Basal–Mean Experimental/Mean Basal*100).

## Ultra-Performance Liquid Chromatography (UPLC) measurements of adenosine nucleotides in whole tissue

UPLC measurements of adenosine nucleotides in whole muscle tissue have been described in detail previously [24]. Briefly, isolated muscles were flash frozen in liquid nitrogen, homogenized in ice-cold perchloric acid using a glass on glass homogenizer, and centrifuged to remove precipitated proteins. Samples were neutralized using potassium hydroxide and centrifuged a second time, to remove perchlorate salt. Adenosine nucleotides and degradation products were assayed using an Acquity UPLC H class system (Waters, Milford, MA). Metabolites were identified by comparison of peak retention times of pure, commercially available standards (Sigma–Aldrich). These UPLC measures can provide an index of intracellular energetic state. The amount of IMP reflects longer periods of metabolic demand exceeding supply as the available adenylate pool is decreased via irreversible deamination of AMP to IMP. Over the timeframe of these stimulation protocols, IMP accumulation is a reliable measure of sustained mismatch between ATP supply and demand. (Adenosine triphosphate-ATP, adenosine diphosphate-ADP, adenosine monophosphate-AMP, and inosine monophosphate-IMP).

## Statistical analysis

Data are represented by mean ± sample standard deviation (SD). Analyses and plotting were carried out using Graphpad prism (V8.01; Windows 10). Two-way analysis of variance (ANOVA) was used for comparison of group means. Assumption of equal variance was tested using a Brown-Forsythe test. Multiple comparisons were tested using Sidak's method. Time series data were compared by fitting the curves for each group with an appropriate regression model (simple linear model for $O_2$ conditions; 3-parameter logistic growth model for $N_2$ conditions). Standard deviation of the residuals was used to determine goodness of fit for all curve fitting. Slopes (linear models) and $P_{50}$ values (i.e. the number of contractions to 50% of the initial force/work; non-linear models) were compared between fed/fasted groups using a sum of squares F-test. P values of < 0.05 were considered statistically significant for all analyses.

## Results

Extensor digitorum longus (EDL) and soleus muscles were chosen for their known differences in thermodynamic efficiency [25]. The muscles also characteristically rely on different modes of energy metabolism for sustained contractions (glycolytic and oxidative energy metabolism respectively) [26]. We used a two-factor two-level experimental design (Fig 1) to test the hypothesis that fasting would significantly reduce the amount of time during hypoxia that the muscles could remain functional.

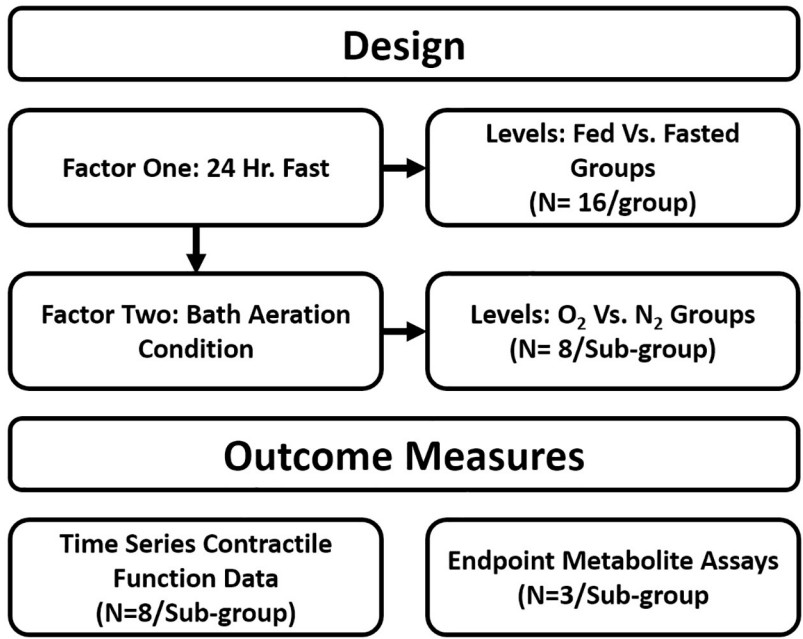

**Fig 1. Experimental design.** Diagram showing the two-factor two-level experimental design implemented in this study. Groupwise comparisons were analyzed using two-way ANOVA. Time series data were analyzed by comparing regression parameters with a sum of squares F-test. P values $\leq.05$ were considered statistically significant. $L_o$ is the optimal resting length of the muscle in mm. $O_2$ and $N_2$ aeration conditions were 95%$O_2$/5%$CO_2$ and 95%$N_2$/5%$CO_2$ respectively in Krebs Ringer solution at ~22°C.

Fasting is a well characterized and effective method of whole body carbohydrate reduction in mice, due to their high thermal conductivity and large surface area to body volume ratio [27]. This method was chosen for this study because it is independent of the confounding effects of exercise or contraction induced fatigue [28]. The mean change in bodyweight over the fasted period (24 hours) was 3.9 ± 0.12 grams, approximately 13% of the mean initial weight. We observed a large difference in stored glycogen levels between fed and fasted groups in both liver (~90% lower) (Table 1) and skeletal muscle (~50% lower) (Table 1). The baseline glycogen concentration was higher in the soleus than the EDL under both fed and fasted conditions. Additionally, soleus muscles had a lower mean glycogen concentration in the fasted group relative to the fed state (mean percent difference of 41.6% compared to 56.1% in the EDL groups; Table 1).

Fasting had no effect on the isometric force-frequency relationship at baseline or under any of the tested conditions in the EDL (Fig 2A) or soleus (Fig 2B), indicating reduced carbohydrate pool size did not alter excitation-contraction coupling. Specific force values for both muscles were consistent with those obtained previously [21]. Additionally, we observed characteristic reductions in maximal specific force following the $O_2$ protocols (and completely impaired force production following the $N_2$ protocols) in both muscles (Fig 2A and 2B). Notably, the isometric force capacity during each protocol did not differ between the fed and fasted groups in either the EDL (Fig 2C) or the soleus (Fig 2D). Similarly, the work capacity over the course of the protocols did not differ for either muscle between the fed and fasted states (Fig 2E and 2F). As expected, the force and work capacities were greatly reduced under the $N_2$ conditions compared to $O_2$ conditions.

Given that no substantial differences in force or work capacities were observed, we next examined whether the timing of muscle functional impairments would differ between the fed

**Table 1. Basal tissue glycogen concentrations in the liver and skeletal muscle of fed Vs. fasted groups.**

| Tissue | Condition | Glycogen (nmol/mg) | StDev (nmol/mg) | % Fed Group |
|---|---|---|---|---|
| Liver | Fed | 387.9* | 88.43 | |
| | Fasted | 42.0* | 20.7 | 10.8 |
| EDL | Fed | 34.4*# | 8.0 | |
| | Fasted | 19.3*# | 3.2 | 56.1 |
| Soleus | Fed | 61.9*# | 17.5 | |
| | Fasted | 25.8*# | 3.6 | 41.6 |

Units are nanomoles hydrolyzed glucosyl units/milligram tissue wet weight (nmol/mg). Data analyzed using two-way ANOVA.

*p <.05 Fed V. Fasted Groups.

#p <.05 EDL V. Sol. N = 4. Sample standard deviation (StDev).

and fasted states. The time-tension integral (TTI) of the isometric portion of each contraction was plotted as a function of the number of contractions (or time) during each protocol for the EDL (Fig 3A) and soleus (Fig 3B). This measurement represents the ability of the muscle to perform sustained non-shortening contractions. Additionally, the length-time integral of the isokinetic portion of each contraction was also plotted against the number of contractions for the EDL (Fig 3C) and soleus (Fig 3D). This measurement represents the ability of the muscle to perform shortening work. Both sets of curves were characterized by an inverse linear relationship under $O_2$ conditions and a distinctly non-linear inverse relationship under $N_2$ conditions during the time and frequency domains of the experiments. The muscles from the fasted groups experienced more rapid reduction in both TTI and work (Fig 3A–3D). Passive tension was measured at the start of each contraction, and the mean maximal values observed during the protocols are reported for the EDL (Fig 3E) and soleus (Fig 3F). This measurement represents stiffening of the muscle, which may be due to several possible factors, including impaired calcium reuptake or cellular swelling due to uncontrolled fluid uptake [29]. None of the muscles experienced substantial changes in passive tension during the $O_2$ protocol. Greater increases in maximal passive tension occurred in the fasted groups (compared to fed groups) in both muscle types under $N_2$ conditions. To account for the possibility that the muscles were accumulating excessive water, the wet weights of the EDL (Fig 3G) and soleus (Fig 3H) were plotted. No differences in wet weight between the fed and fasted states were observed in either muscle and all the tested muscles accumulated additional weight following the $N_2$ protocol.

We next measured the muscle glycogen levels following the $O_2$ and $N_2$ protocols. The $N_2$ protocol reduced glycogen concentrations in all the muscles tested, relative to the $O_2$ condition (Table 2). Additionally, glycogen concentrations were lower in the fasted soleus groups compared to the fed groups under both $O_2$ and $N_2$ conditions (Table 2). However, glycogen concentrations did not differ between fed and fasted groups in the EDL muscles. Using the response coefficient ($R_{Glyc}$), allowed for comparison of each group mean to the basal values that are presented in Table 1. The patterns among both muscle types were similar when represented this way. The largest differences observed were between $O_2$ and $N_2$ conditions and were not substantially different between fed and fasted groups. Interestingly, the smallest difference in glycogen concentration observed was in the fasted $O_2$ condition for each muscle. This observation likely indicates the use of alternative (oxygen dependent) fuel sources.

Total adenosine nucleotide (TAN) concentrations were examined as a measure of the aggregate tissue energetic state at baseline and at the end of each protocol. Reductions in the concentrations of the total adenosine nucleotide pool, and accumulation of IMP, are measures of the muscles' inability to resynthesize ATP [24]. Hypoxia resulted in statistically significant

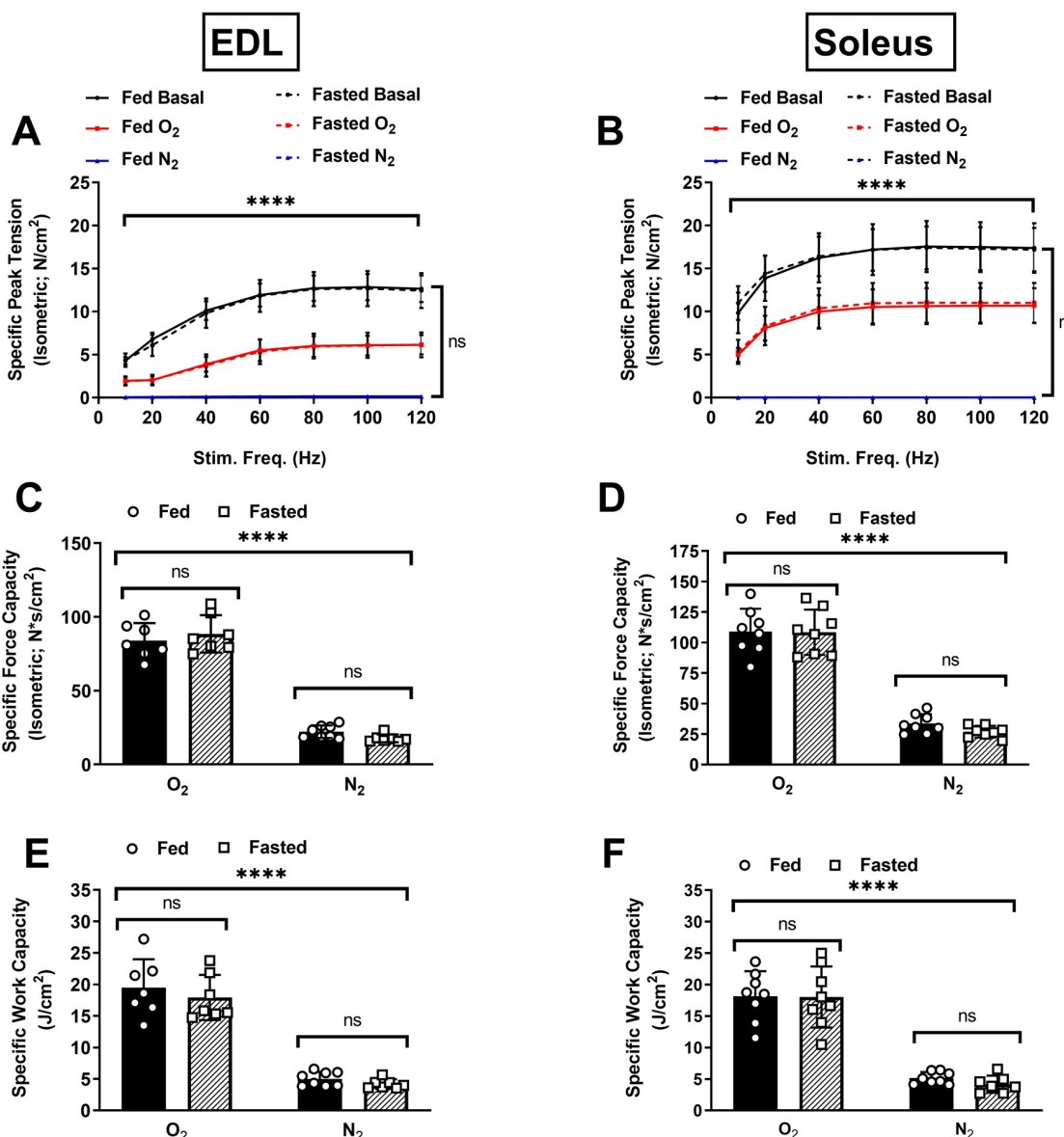

**Fig 2. Effects of carbohydrate depletion on excitation-contraction coupling and force/work capacities.** Specific force-frequency curves for EDL (A) and soleus (B). Basal conditions are 95% $O_2$ prior to isokinetic protocol. (C,D) Specific force capacities were obtained by summing the isometric portion of the time-tension integrals at each sampling interval for the EDL and soleus respectively. (E,F) Specific work capacities were obtained by summing the isovelocity (shortening) portion of the length-tension integrals at each sampling interval for the EDL and soleus. Solid black bars = Fed Group. Crosshatched bars = Fasted Group. N = 8/ treatment/group (EDL), N = 7/treatment/group soleus. Data are presented as mean ± SD. Group means were compared using two-way ANOVA. ****p <.0001 statistically significant effect of bath aeration condition. ns = no significant effect of feeding condition.

reduction in the all the adenosine nucleotides measured, except for adenosine monophosphate (AMP) in the soleus (Fig 4A–4F). A significant increase in the AMP degradation product inosine monophosphate (IMP) following hypoxia was observed in all groups tested (Fig 4G and 4H). This observation supports the notion that the muscles were in a state of energetic failure at the completion of the protocols. No differences were observed between the fed and fasted groups in either muscle type for any of the measured nucleotides (Fig 4A–4H).

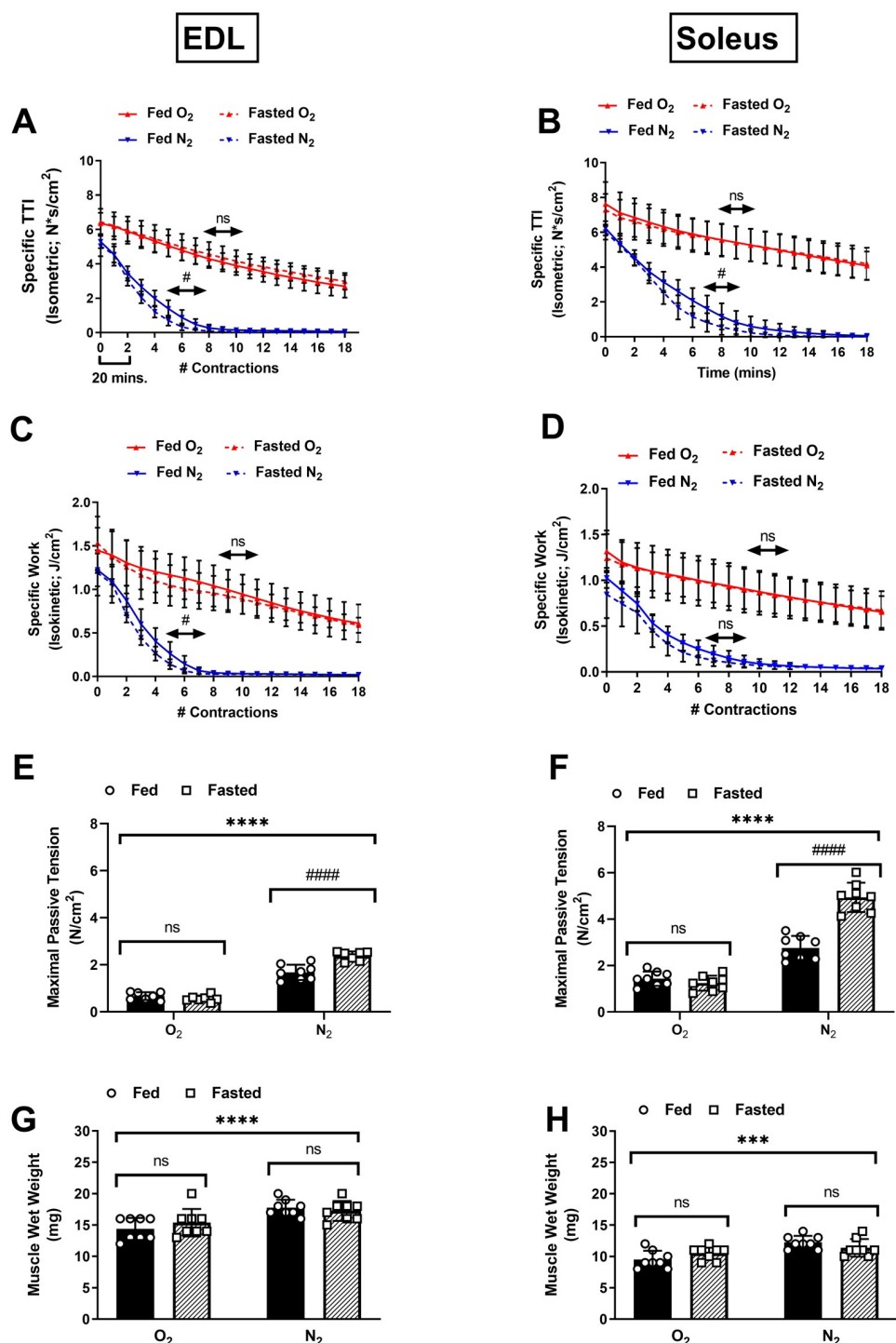

**Fig 3. Effects of carbohydrate depletion on the timing of functional impairment during hypoxia and nutrient deprivation (HND).** Isometric time-tension integrals (TTI) of each contraction over the course of 18 contractions (or 180 minutes) under each condition for the EDL (A) and soleus (B). Isokinetic length-tension integrals (isokinetic work) of each contraction for the EDL (C) and soleus (D). Maximal passive tension developed during the protocol (measured at the start of each contraction) for the EDL (E) and soleus (F). Muscle wet weights obtained at the end of each protocol for the EDL (G) and soleus (H). N = 8/treatment/group (EDL), N = 7/treatment/group soleus. Data are presented as mean ± SD. For the $O_2$ condition data, the slope of each line was determined using a simple linear regression model. For $N_2$ data, the number of contractions to 50% initial force/work ($P_{50}$) was estimated using non-linear regression. Parameter values (A-D) were compared using a sum of squares F-test. Solid black bars = Fed Group.

Crosshatched bars = Fasted Group. Group means (E-H) were compared using two-way ANOVA. ****p <.0001 statistically significant effect of bath aeration condition. ns = no significant effect of feeding condition. # p <.05 statistically significant effect of feeding condition; #### p <.0001 statistically significant effect of feeding condition (Sidak's multiple comparison test).

Previous reports have indicated that dystrophin IF staining is rapidly reduced in skeletal and cardiac muscle during early myonecrosis [19,30]. Immunofluorescent staining for the sarcolemmal protein dystrophin and the extracellular matrix protein laminin was performed on a subset of transverse sectioned muscles to assess the possibility that muscles were incurring damage during the contraction protocols. No apparent changes were observed in the EDL (Fig 5A) or soleus (Fig 5C) under $O_2$ or $N_2$ conditions, indicating that the muscle tissue remained intact during the experiments. Degradation of myofibrillar structures are another well characterized indicator of myonecrosis development [31]. Parallel assessments were made to accompany the dystrophin/laminin stain. Fibrous actin was stained in fixed whole mount muscle specimens utilizing optical sectioning to assess the intramyofibrillar-IMF and perinuclear-PN regions of the myofibers at baseline and following the $N_2$ protocol in the EDL (Fig 5B) and soleus (Fig 5D). Together these qualitative assessments did not reveal any indication of damage.

## Discussion

Skeletal muscle is among the most metabolically dynamic tissues in the body, and is capable of sustaining a 100-fold change in ATP utilization rate during contraction [32]. The total cost of ATP during contraction is proportional to the duration, intensity, and type (i.e. shortening vs. non-shortening) [33]. Glycogen is the primary storage form of glucose in skeletal muscle, and is a major source of fuel during most forms of muscle activity [34].

Importantly, glycogen is also the primary source of stored fuel utilized to regenerate ATP via substrate level phosphorylation in anaerobic glycolysis during severe hypoxia [33]. Depletion of stored muscle glycogen by fasting or exhaustive exercise results in impaired fatigue resistance and recovery in isolated rodent muscles under normoxic conditions [34–36]. Under hypoxic conditions, this effect would be expected to lead to cumulative reductions in energetic capacity due to the inability to resynthesize ATP and phospho-creatine (PCr) that is used in support of contraction or resting metabolic processes.

**Table 2. Tissue glycogen concentrations in EDL and soleus muscles of fed V. fasted mice following $O_2$ or $N_2$ protocols.**

| Tissue | Group | Condition | Glycogen(nmol/mg) | StDev(nmol/mg) | $R_{Glyc}$ (%) |
|---|---|---|---|---|---|
| EDL | Fed | $O_2$ | 22.4† | 4.4 | -34.8 |
| | | $N_2$ | 7.9† | 5.1 | -77.0 |
| | Fasted | $O_2$ | 18.8† | 7.3 | -2.5 |
| | | $N_2$ | 7.6† | 1.3 | -60.6 |
| Soleus | Fed | $O_2$ | 42.1†* | 6.6 | -31.9 |
| | | $N_2$ | 29.0†* | 1.6 | -53.1 |
| | Fasted | $O_2$ | 23.5†* | 8.8 | -8.9 |
| | | $N_2$ | 9.8†* | 4.2 | -62.0 |

Units are nanomoles hydrolyzed glucosyl units/milligram tissue wet weight (nmol/mg). The Response Coefficient ($R_{Glyc}$) indicates the percent change relative to the baseline group means (Presented in Table 1). Sample standard deviation (StDev). Group means were compared using two-way ANOVA.

*p <.05 Fed V. Fasted Groups.

†p <.05 $O_2$ V. $N_2$ Groups. N = 4/group.

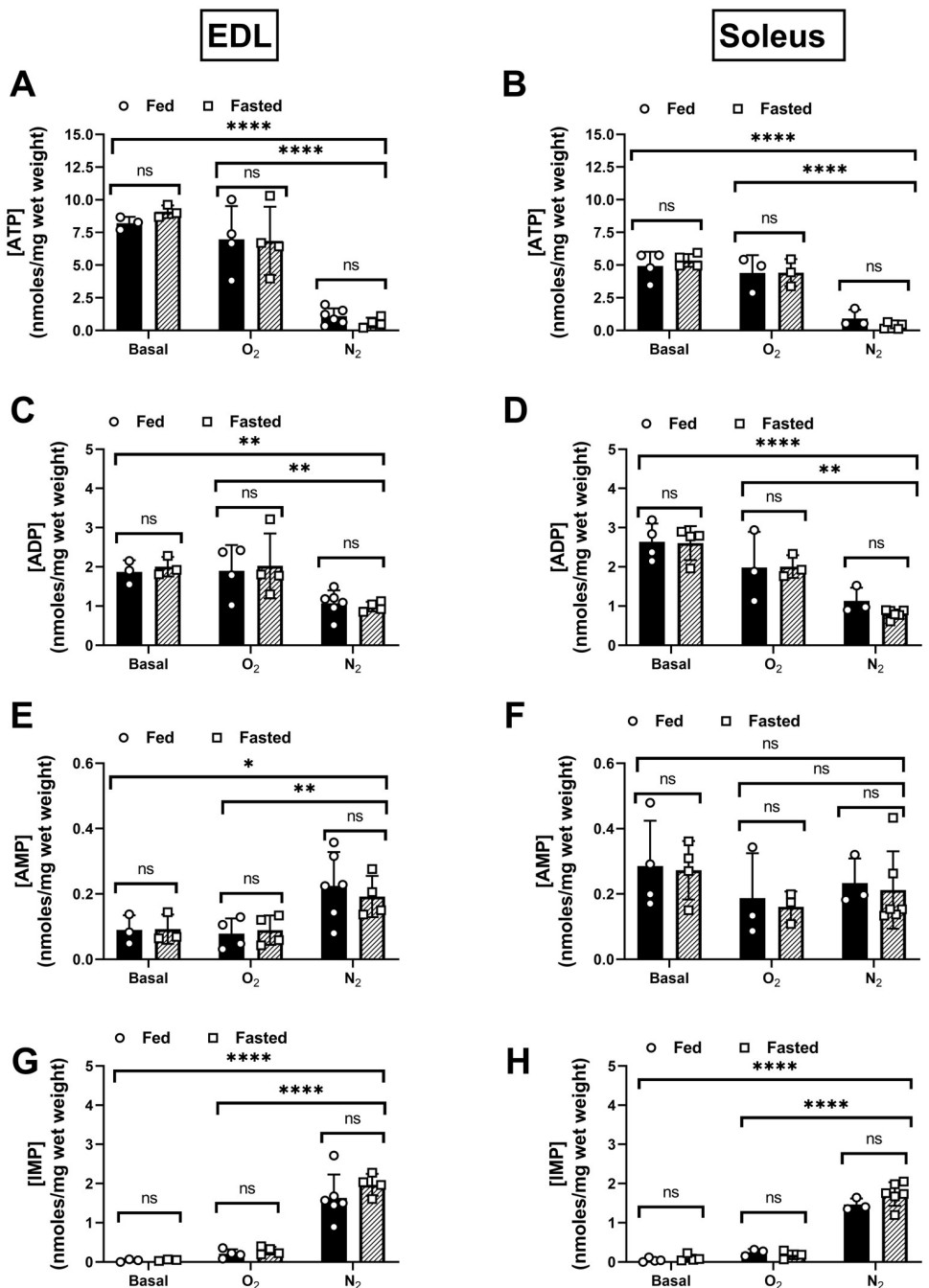

**Fig 4. Adenosine nucleotide profiles of fed/fasted EDL and soleus muscles at baseline and following 180 min. $O_2$ or $N_2$ protocols.** Whole muscle tissue adenosine triphosphate (ATP) concentrations at baseline and after the $O_2$ and $N_2$ protocols for the EDL (A) and soleus (B). Adenosine diphosphate (ADP) concentrations for the EDL (C) and soleus (D). Adenosine monophosphate (AMP) concentrations for the EDL (E) and soleus (F). Inosine monophosphate (IMP) concentrations for the EDL (G) and soleus (H). Solid black bars = Fed Group. Crosshatched bars = Fasted Group. N = 3/treatment/group. Data are presented as mean ± SD. Group means were compared using a two-way ANOVA. ****p <.0001, **p <.005, *p <.05 statistically significant effect of bath aeration condition. ns = no significant effect of feeding condition.

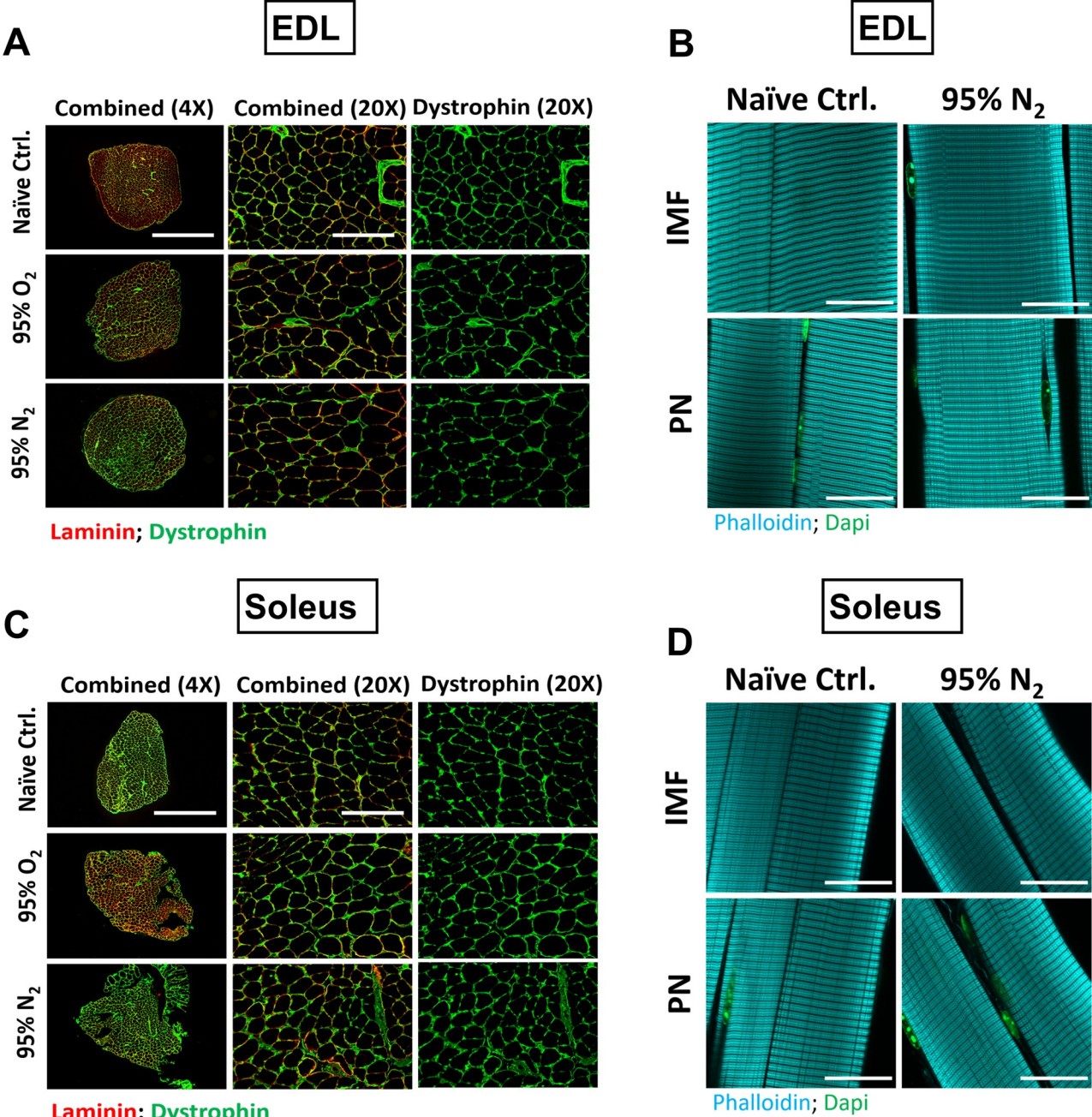

**Fig 5. Qualitative assessment of structural integrity of the muscles following experimental protocols.** To control for the possibility that the muscles were structurally damaged during the contraction protocols, we performed immunofluorescence against sarcolemmal and extracellular matrix proteins. Image panels of dystrophin (green), and laminin (red) stained transverse EDL (A) and soleus (C) muscle sections under each of the conditions tested. Sarcomeric actin was stained using phalloidin (Cyan) in fixed/permeabilized whole mount muscles at baseline or following 180 mins of severe hypoxia (95% $N_2$); EDL (B) and soleus (D). Optical sectioning facilitated imaging in the intra-myofibrillar (IMF) and perinuclear (PN) regions of the muscle fibers. Scale bars are 1000μm (A, B Left Panel), 200 μm (A,B right panels), and 25 μm (B,D). N = 1/timepoint.

Overnight fasting in rodents results in more dramatic metabolic effects than human overnight fasting, but induces experimentally reproduceable reductions in systemic carbohydrate stores that are similar to more extreme physiological conditions such as hyperinsulinemia, hypoglycemia, or post exercise recovery [11,14,27,37]. Fasting was used in this study because it

is independent of the confounding effects of exercise or contraction induced fatigue [28]. Notably, the glycogen values observed in this study differ from several previous reports in that fed state control values for both muscles are relatively high, and that the soleus glycogen levels are substantially higher than the EDL (values between muscles did not differ in the previous reports) [38–40].

Muscle glycogen concentration is a physiologically dynamic parameter that is influenced by experimental conditions such as assay method and normalization factor, as well as biological conditions such as parental genetic background and metabolic state [41–43]. Though we cannot directly account for specific confounders that explain the discrepancy in this study, there are two likely candidates that should be considered for further investigation: 1.) Muscle glycogen levels have been shown to vary with season and diurnal cycle in mice [44] and rats [45,46], with peaks in the dark-light cycle transition period (the time at which animals were sacrificed in this study). 2.) Soleus muscles have been shown to be more sensitive to insulin stimulated glucose uptake and glycogen synthesis in both mice [38] and rats [47], and insulin stimulated glucoregulatory responses have been shown to differ among inbred mouse strains [48]. Taken together, the described findings support the possibility that stored muscle glycogen values may have been influenced by seasonal, circadian, or hormonal variation intrinsic to the genetic background of the mice used in this study.

We were somewhat surprised to find that fasting associated reductions in muscle glycogen levels had no substantial effect on the timing or magnitude of isolated muscle functional impairment under hypoxic conditions. Our findings indicate that both muscle types retain a large pool of stored glycogen that is non-essential for reserve mechanical force production during hypoxia. It is not clear what the reserve glycogen pool contributes to *in vivo* during fasting. Future studies could be directed to investigate its' potential involvement in the maintenance of systemic glucose homeostasis through the production of free amino acids (i.e. alanine and glutamine) or lactate which can be converted to glucose in the liver [49,50].

In mouse EDL and soleus muscles, as much as 50% of the resting metabolic rate has been attributed to maintenance of intracellular calcium homeostasis [51]. We observed greater maximal passive tension development in the fasted group relative to the fed group under $N_2$ conditions in both muscle types. This phenomenon is most likely indicative of progressive impairment of calcium handling as the capacity for ATP re-synthesis was gradually depleted [24]. This effect may have implications for reperfusion timing, as it has been noted that calcium handling impairment prior to reperfusion is associated with poor salvage outcomes [4,52].

Skeletal muscle fiber types are categorized by a range of intrinsic metabolic and mechanical properties [53]. Human muscles are generally of mixed fiber type, but mouse muscles consist of more homogeneous fiber type distributions, making them a practical model for studying fiber type specific effects (soleus: 1:1 slow type I/fast type IIa; EDL: 9:1 fast type IIb/fast type IIa) [26,54]. At face value, it may seem intuitive that fast glycolytic fiber types would be better suited to performance during hypoxia due to their preference for stored carbohydrate dependent energy metabolism [34,35,55]. However, several studies have indicated a high degree of sensitivity of fast glycolytic muscles to ischemia/reperfusion injury [16,17,56]. One important contributing factor to this effect is an energetic inefficiency of contraction due to interactions at the level of the acto-myosin crossbridges [25,57].

In this study, we observed that Soleus muscles stored more glycogen at baseline, had greater specific force/work capacities, and produced absolute force for a longer period during hypoxia compared to EDL muscles. The observations regarding greater glycogen content in the soleus muscle compared to EDL muscles are not consistent with previous reports [39,40,58], but the observations of improved mechanical function during hypoxia in soleus compared to EDL muscles have been previously reported using small muscles isolated from rats [56]. Though the

absolute differences in glycogen concentrations between groups were larger in the soleus compared to the EDL, the response coefficient ($R_{Glyc}$) which facilitates interpretation of group differences relative to their baseline concentration, indicated that the patterns of utilization were not different between the two types of muscles. We interpret these findings to mean that the greater basal glycogen concentration observed in the soleus muscles was likely not the primary factor underlying it's enhanced ischemic mechanical performance.

There are several important limitations to this study. First, the carbohydrate reduction associated with fasting is not complete, leaving approximately half of the fed state muscle glycogen available during experimental hypoxia. Though this is independent of confounding effects associated with other methods of glycogen depletion [28], there are other effects of fasting that may confound observed outcomes [27]. Second, though we were able to assess muscle contractile function in time series, we were only able to assess changes in muscle metabolite levels (e.g. adenine nucleotides, glycogen, etc.) at baseline and after the 180-minute protocols. Additional experimentation is necessary to fully characterize the time-dependent changes in key metabolites during hypoxia. Finally, our experimental system utilizing isolated muscle is highly controlled for the effects of hypoxia but lacks the biological variability and complexity of ischemia. We hope that the observations in this study can be used to inform development of hypotheses that can be further tested using *in vivo* preclinical models of ischemia.

## Conclusion

Identification of key factors that affect the timing of muscle energetic failure during hypoxia will aid in identifying optimal windows for therapeutic intervention in ischemic disease. We predicted that the amount of stored carbohydrate is one such factor, as it is a major contributor to anaerobic energy metabolism and is influenced by several physiologically relevant conditions. We conclude that mouse hindlimb muscles maintain a large pool of stored carbohydrate that is utilized during fasting but does not contribute substantially to the timing of functional decline during acute hypoxia. The carbohydrate lowering effects associated with fasting did not substantially affect the total capacity or timing of contractile function impairment in either muscle type. However, fasting did result in substantial increases in passive tension development during hypoxia, which may have implications for the design of follow up studies *in vivo*. We also found that soleus muscles maintained a greater total force capacity and became impaired more slowly than EDL muscles, independent of glycogen utilization during the experimental period. This finding supports several previous observations and bolsters the notion that susceptibility to hypoxia-induced impairment is not uniform across muscle types.

## Author Contributions

**Conceptualization:** Cameron A. Schmidt, Emma J. Goldberg, Tom D. Green, Reema R. Karnekar, Jeffrey J. Brault, Spencer G. Miller, Adam J. Amorese, Dean J. Yamaguchi, Espen E. Spangenburg, Joseph M. McClung.

**Data curation:** Cameron A. Schmidt, Emma J. Goldberg, Tom D. Green, Reema R. Karnekar, Jeffrey J. Brault, Spencer G. Miller, Adam J. Amorese, Dean J. Yamaguchi, Espen E. Spangenburg, Joseph M. McClung.

**Formal analysis:** Cameron A. Schmidt, Emma J. Goldberg, Tom D. Green, Reema R. Karnekar, Jeffrey J. Brault, Spencer G. Miller, Adam J. Amorese, Dean J. Yamaguchi, Espen E. Spangenburg, Joseph M. McClung.

**Funding acquisition:** Jeffrey J. Brault, Espen E. Spangenburg, Joseph M. McClung.

**Investigation:** Cameron A. Schmidt, Emma J. Goldberg, Tom D. Green, Reema R. Karnekar, Jeffrey J. Brault, Spencer G. Miller, Adam J. Amorese, Dean J. Yamaguchi, Espen E. Spangenburg, Joseph M. McClung.

**Methodology:** Cameron A. Schmidt, Joseph M. McClung.

**Project administration:** Cameron A. Schmidt, Joseph M. McClung.

**Resources:** Cameron A. Schmidt, Joseph M. McClung.

**Validation:** Cameron A. Schmidt.

**Visualization:** Cameron A. Schmidt, Joseph M. McClung.

**Writing – original draft:** Cameron A. Schmidt.

**Writing – review & editing:** Cameron A. Schmidt, Joseph M. McClung.

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
