## [Decision Letter · Decision Letter 0]

18 Dec 2019

PONE-D-19-31637

Effects of fasting induced carbohydrate depletion on murine ischemic skeletal muscle function.

PLOS ONE

Dear Dr. McClung,

Thank you for submitting your manuscript to PLOS ONE. After careful consideration, we feel that it has merit but does not fully meet PLOS ONE’s publication criteria as it currently stands. Therefore, we invite you to submit a revised version of the manuscript that addresses the points raised during the review process.

The reviews have found merit in your study however, they have both expressed concerns about the presentation of the data which would need to be addressed to make the manuscript suitable for publication. In particular the applicability of your model to ischemia needs to be addressed. It is also important to reconcile the reported muscle glycogen concentrations with those commonly reported in the literature. 

We would appreciate receiving your revised manuscript by Feb 01 2020 11:59PM. To enhance the reproducibility of your results, we recommend that if applicable you deposit your laboratory protocols in protocols.io, where a protocol can be assigned its own identifier (DOI) such that it can be cited independently in the future. For instructions see: http://journals.plos.org/plosone/s/submission-guidelines#loc-laboratory-protocols

We look forward to receiving your revised manuscript.

Kind regards,

Cameron J. Mitchell, PhD

Academic Editor

PLOS ONE

Journal Requirements:

Please ensure that your manuscript meets PLOS ONE's style requirements, including those for file naming. The PLOS ONE style templates can be found at http://www.plosone.org/attachments/PLOSOne_formatting_sample_main_body.pdf and http://www.plosone.org/attachments/PLOSOne_formatting_sample_title_authors_affiliations.pdf We note that you have stated that you will provide repository information for your data at acceptance. Should your manuscript be accepted for publication, we will hold it until you provide the relevant accession numbers or DOIs necessary to access your data. If you wish to make changes to your Data Availability statement, please describe these changes in your cover letter and we will update your Data Availability statement to reflect the information you provide. PLOS requires an ORCID iD for the corresponding author in Editorial Manager on papers submitted after December 6th, 2016. Please ensure that you have an ORCID iD and that it is validated in Editorial Manager. To do this, go to ‘Update my Information’ (in the upper left-hand corner of the main menu), and click on the Fetch/Validate link next to the ORCID field. This will take you to the ORCID site and allow you to create a new iD or authenticate a pre-existing iD in Editorial Manager. Please see the following video for instructions on linking an ORCID iD to your Editorial Manager account: https://www.youtube.com/watch?v=_xcclfuvtxQ

Additional Editor Comments (if provided):

The bar graphs used would be much more informative if individual data points were overlaid to show the true variability in the data. 

Reviewers' comments:

Reviewer's Responses to Questions

**Comments to the Author**

1. Is the manuscript technically sound, and do the data support the conclusions?

Reviewer #1: Partly

Reviewer #2: Partly

2. Has the statistical analysis been performed appropriately and rigorously? 

Reviewer #1: No

Reviewer #2: I Don't Know

3. Have the authors made all data underlying the findings in their manuscript fully available?

Reviewer #1: Yes

Reviewer #2: Yes

4. Is the manuscript presented in an intelligible fashion and written in standard English?

Reviewer #1: Yes

Reviewer #2: Yes

5. Review Comments to the Author

Reviewer #1: The main idea of this manuscript is investigating the role of muscle glycogen amount for ischemic muscle function. The authors hypothesized that reduced muscle glycogen would shorten the functional time of ischemic muscle. To test this hypothesis, the authors used fasting to reduce glycogen storage and measured muscle functions with hypoxia condition in vitro. Although the idea was interesting, reduced muscle glycogen did not likely change the muscle functions with/without ischemia (rejected hypothesis). I recommend several changes as below to improve the manuscript.

Fig 1. Need quantification for each image. Vessel density should be normalized by fiber area. Without quantification, this figure can be a supplemental figure. Analyzed fiber number and total animal number should be added to the legend. Make more connections between figure 1 and rest of figures. Otherwise, this figure is little bit out of place.

Fig 2. To help readers better understand this and later figures, Please add scheme (cartoon) of experiment, which will help to understand this and later figures. Also add statistical analysis method for this figure to the legend. Since data have more than 2 factors (O2 vs N2 and Fed vs. Fasting), 2-way ANOVA should be used. (Please note, the Methods section only mentioned a 2 tail student test for group comparison) (Minor comments for Fig 2C-F, adding crosshatch as a fill for the fasting graph (but keeping the color) would make the graph easier to read.)

Fig 3. Add statistical analysis (such as * marks) for each figure and add statistical analysis method (2 way ANOVA is recommended) to the legend. (Same minor comments for Fig 3G-H, adding crosshatch to the fasting graph would improve clarity of the image)

Fig 4. Consider change the way the figure is presented. It would be nice combine ATP, ADP, AMP values of all conditions in each single graph like Fig 4G-H. Also add statistical analysis in graph and methods to the legend.

Fig 5. Add quantification graph for image analysis, including number of sections and number of mice with statistical analysis method.

Result. Line 319-320. The author claims that “the muscles from the fasted group experienced more rapid reduction in both TTI and work”. However, Fig 3c and 3d did not support this claim due to lack of statistical analysis. Error bars of fed and fasted group graph seem to overlap each other at almost every time point. Please provide detailed support/explanation for this claim.

Reviewer #2: This manuscript studies the effects of fasting and hypoxia in ex vivo soleus and EDL mouse muscles that are stimulated to contract. The measurements include contractile function, passive tension and metabolites such as glycogen, IMP, and TAN. The premise is that the hypoxia, which is produced by incubated muscles in solution gassed with N2, is a model for ischemia. This premise if flawed. Fasting is used as an intervention to reduce tissue carbohydrate supply. Fasting does lower carbohydrate content of muscles, but that is not the only thing that fasting does. The text needs to be revised to use more direct and accurate language to describe the experimental approach. The major metabolic measurement is glycogen, and some of the values disagree with literature values, which undermines confidence in the data.

MAJOR

The title is misleading and should be revised to something more accurate, such as “Effects of fasting and ex vivo hypoxia on murine skeletal muscle contractile function.” The authors don’t have to use this exact wording, but “ischemia” should not be used, “hypoxia” should be used, and the function should be specified.

Ischemia refers to low blood flow. The muscles are studied ex vivo without any flow, whether they are oxygenated or not. Low oxygen is not the only consequence of ischemia. There is no convincing evidence that this is a good model for ischemia. The repeated use of the word “ischemia” or “ischemic” to describe the experiment should be eliminated. The experiment is studying hypoxia, not ischemia, and the text in the entire manuscript should be revised accordingly. Eliminating or at least deemphasizing the assertion that the experimental approach is an ischemia model would be helpful. If the authors are determined to comment on how this model has relevance to ischemia, they need to provide specific and direct evidence to support this assertion, and to also directly acknowledge the limitations of this experimental approach as an ischemia model.

The abstract refers to “conditions of reduced carbohydrate supply” before using a more informative description of “fasting.” The abstract should identify the duration of fasting. The text throughout should also not suggest that all fasting does is reduce carbohydrate supply or glycogen levels. It is OK to indicate that this might be an important consequence of fasting for the effects on contraction function, but there should be a more accurate description of what fasting represents and recognition that reducing glycogen is not everything it does.

A major point made by the authors is that glycogen concentration is much higher in the soleus than the EDL. This result has not been observed in earlier research. Glycogen of soleus was not much greater for mouse soleus compared to EDL (Jorgensen J Biol Chem. 2004. 279(2):1070-9; Bonen J Appl Physiol 1994. 76(4):1753-8). The authors should address what might account for the discrepant results and provide evidence that their results are consistent with results of a number of earlier studies.

The muscle glycogen concentrations are higher than usually reported for mouse EDL and soleus. The value in Table 1 for fed soleus (61.9 nmol/mg) is very high compared to the literature. There should be citations of literature values for glycogen and an explanation for the high values in this study compared to the literature.

The light/dark cycle times should be stated, and the times when fasting began and when muscles were sampled should be stated.

The Methods section (lines 148-150) on fasting refers to a pilot study and cites a study (ref 23) that is not from this group of authors. It is confusing to know if the authors performed a pilot study or not, and why they cited this study.

In the statistics section (lines 242-243), it stated that both SEM and SD are used with the data. Either one or the other should be used. SD is more informative.

The Discussion should acknowledge important limitations of the study. One would be that only one timepoint was studied for metabolite concentrations. Measurements at several timepoints would make the study more informative.

The final sentence of the Introduction is that the “This information …ischemia models.” The Conclusion states that the results are valuable for therapeutic intervention (lines 479 and 488). It is unclear why this information will be valuable for either these ischemia models or for therapeutic interventions. It should be directly stated why this information will be valuable.

Figure 4 should include text on the figure itself to indicate which A-F panels are from the O2 treatment and which are from N2 treatment.

In the Introduction (line 76), it stated that the experiment was intended to determine the “exact temporal nature…”, but only one time point (3 hours) was studied, so this study doesn’t determine the “exact temporal nature” of the results.

6. PLOS authors have the option to publish the peer review history of their article (what does this mean?). If published, this will include your full peer review and any attached files.

Reviewer #1: No

Reviewer #2: No

---

## [Author Response · Author response to Decision Letter 0]

28 Jan 2020

Additional Editor Comments:

The bar graphs used would be much more informative if individual data points were overlaid to show the true variability in the data. 

-We thank the editor for this suggestion and agree that the recommended change would improve the manuscript. We have modified all bar graphs accordingly. Figures: 2C-F; 3G-H; and 4A-H. 

Review Comments to the Author

Reviewer #1: The main idea of this manuscript is investigating the role of muscle glycogen amount for ischemic muscle function. The authors hypothesized that reduced muscle glycogen would shorten the functional time of ischemic muscle. To test this hypothesis, the authors used fasting to reduce glycogen storage and measured muscle functions with hypoxia condition in vitro. Although the idea was interesting, reduced muscle glycogen did not likely change the muscle functions with/without ischemia (rejected hypothesis). I recommend several changes as below to improve the manuscript.

-We thank the reviewer for their time and for helping us improve the quality of this manuscript with constructive feedback. We have addressed each comment below and summarized the changes made below each comment. 

Fig 1. Need quantification for each image. Vessel density should be normalized by fiber area. Without quantification, this figure can be a supplemental figure. Analyzed fiber number and total animal number should be added to the legend. Make more connections between figure 1 and rest of figures. Otherwise, this figure is little bit out of place.

-We agree with this assessment and thank the reviewer for this suggestion. In order to keep the manuscript message focused, we have removed the original Figure 1 and replaced it with a schematic diagram that summarizes the experiments (Also addressed in the second comment). We have also removed the paragraph that referred to the original figure 1 from the discussion: 

“The dynamic requirements of ATP during muscle contraction require a similarly dynamic supply of carbon fuel sources that are derived from both blood and intracellular stores. Physiological and anatomical adaptations (i.e. capillary density, size of stored energy substrate pools, and mitochondrial density/function) are known to facilitate large differences in capacity for spontaneous vs. sustained exercise in different species(44). Similar adaptive differences are highlighted in the distinct anatomical, mechanical, and thermodynamic properties of mouse locomotory EDL and postural soleus muscles(23,27,45,46). These adaptive differences, combined with their similar size and relatively homogeneous fiber type compositions(28), make these muscles excellent candidates for comparative studies of muscle energy metabolism.”

Fig 2. To help readers better understand this and later figures, Please add scheme (cartoon) of experiment, which will help to understand this and later figures. Also add statistical analysis method for this figure to the legend. Since data have more than 2 factors (O2 vs N2 and Fed vs. Fasting), 2-way ANOVA should be used. (Please note, the Methods section only mentioned a 2 tail student test for group comparison) (Minor comments for Fig 2C-F, adding crosshatch as a fill for the fasting graph (but keeping the color) would make the graph easier to read.)

-We apologize for this omission. We have updated the statistical analysis methods section to include details of the 2-way ANOVA. We have updated the methods section: 

Lines: 238-248 “Data are represented by mean ± sample standard deviation (SD). Analyses and plotting were carried out using Graphpad prism (V8.01; Windows 10). Two-way analysis of variance (ANOVA) was used for comparison of group means. Assumption of equal variance was tested using a Brown-Forsythe test. Multiple comparisons were tested using Sidak’s method. Time series data were compared by fitting the curves for each group with an appropriate regression model (simple linear model for O2 conditions; 3-parameter logistic growth model for N2 conditions). Standard deviation of the residuals was used to determine goodness of fit for all curve fitting. Slopes (linear models) and P50 values (i.e. the number of contractions to 50% of the initial force/work; non-linear models) were compared between fed/fasted groups using a sum of squares F-test. P values of < 0.05 were considered statistically significant for all analyses”.

Additionally, we have added crosshatch patterns to the fasted group in the bar graphs as requested. Additionally, we have included the individual data points on the bar graphs to more clearly represent the sample variability. 

Fig 3. Add statistical analysis (such as * marks) for each figure and add statistical analysis method (2 way ANOVA is recommended) to the legend. (Same minor comments for Fig 3G-H, adding crosshatch to the fasting graph would improve clarity of the image)

-We have updated the legends in all figure panels to include details of the relevant statistical analysis. For example: Figure 3 legend: 

Lines 340-349 “N=8/treatment/group (EDL), N=7/treatment/group soleus. Data are presented as mean ± SD. For the O2 condition data, the slope of each line was determined using a simple linear regression model. For N2 data, the number of contractions to 50% initial force/work (P50) was estimated using non-linear regression. Parameter values (A-D) were compared using a sum of squares F-test. Solid black bars = Fed Group. Crosshatched bars = Fasted Group. Group means (E-H) were compared using two-way ANOVA. ****p<.0001 statistically significant effect of bath aeration condition. ns = no significant effect of feeding condition. # p<.05 statistically significant effect of feeding condition; #### p<.0001 statistically significant effect of feeding condition (Sidak’s multiple comparison test)”. 

Fig 4. Consider change the way the figure is presented. It would be nice combine ATP, ADP, AMP values of all conditions in each single graph like Fig 4G-H. Also add statistical analysis in graph and methods to the legend.

-We agree that the recommended changes would improve the interpretation of the figure. We have reorganized the graphs in Figure 4 accordingly. Additionally, we have added individual data points to the bar graphs in the manuscript to better represent sample variability. Finally, we have updated the figure legend to include description of the statistical analysis and relevant symbols indicating statistical significance to the graphs. 

Fig 5. Add quantification graph for image analysis, including number of sections and number of mice with statistical analysis method.

-We understand the reviewer’s concern that the images are not quantified. The images in Figure 5 are intended to demonstrate that muscle is not becoming significantly damaged during the experiment. The images were intended to be qualitative, but thorough in that they demonstrate a lack of damage using multiple histological methods. To address this concern, we have updated the figure legend to highlight the qualitative nature of the images as well as the number of muscles used:

Lines: 409-410, 418 “Fig 5: Qualitative assessment of structural integrity of the muscles following experimental protocols.” “N=1/timepoint”). 

Additionally, we have clarified this point in the results to avoid any confusion: 

Lines: 406-407 “Together these qualitative assessments did not reveal any indication of damage”). 

Result. Line 319-320. The author claims that “the muscles from the fasted group experienced more rapid reduction in both TTI and work”. However, Fig 3c and 3d did not support this claim due to lack of statistical analysis. Error bars of fed and fasted group graph seem to overlap each other at almost every time point. Please provide detailed support/explanation for this claim.

-Excellent point, we apologize for any apparent over-interpretation of the data. We have added additional analysis to include a more direct measure of the rate of muscle mechanical failure for statistical comparison. In brief, we performed linear regression for O2 data sets and non-linear regression for N2 data sets. We selected a three-parameter logistic growth model for the N2 data sets, because it gave a similar fit (determined by standard deviation of the residuals) compared to a four-parameter model but required fewer model assumptions. We then compared curve parameters between groups using a sum of squares F-test. The parameters were: slope (for O2 groups) and P50 (for N2 groups; P50= number of contractions or time to reach 50% of the initial force/work value predicted by the model). 

Please note that these additional analyses did not result in any major changes to our previous interpretation of the data. In summary, our conclusion is that fasted groups did reach 50% of initial force/work faster than fed groups. However, the effect of feeding condition was negligible from a practical perspective because it only separated the groups by a matter of minutes. 

 We have highlighted our interpretation of our observations in the results: 

Lines 316-320 “Both sets of curves were characterized by an inverse linear relationship under O2 conditions and a distinctly non-linear inverse relationship under N2 conditions during the time and frequency domains of the experiments. The muscles from the fasted group experienced more rapid reduction in both TTI and work (Fig A-D)”. 

We have updated the methods section to include description of the additional analysis:

Lines 242-248 “Time series data were compared by fitting the curves for each group with an appropriate regression model (simple linear model for O2 conditions; 3-parameter logistic growth model for N2 conditions). Standard deviation of the residuals was used to determine goodness of fit for all curve fitting. Slopes (linear models) and P50 (i.e. the number of contractions to 50% of the initial force/work; non-linear models) were compared using a sum of squares F-test.”. We have updated Figure 3, the legend, and the methods section to reflect these changes. 

Reviewer #2: This manuscript studies the effects of fasting and hypoxia in ex vivo soleus and EDL mouse muscles that are stimulated to contract. The measurements include contractile function, passive tension and metabolites such as glycogen, IMP, and TAN. The premise is that the hypoxia, which is produced by incubated muscles in solution gassed with N2, is a model for ischemia. This premise if flawed. Fasting is used as an intervention to reduce tissue carbohydrate supply. Fasting does lower carbohydrate content of muscles, but that is not the only thing that fasting does. The text needs to be revised to use more direct and accurate language to describe the experimental approach. The major metabolic measurement is glycogen, and some of the values disagree with literature values, which undermines confidence in the data.

-We thank the reviewer for their time and careful consideration of our manuscript. The provided comments have been very helpful in improving the quality of the work. We have provided individual responses to the comments below, and included brief summaries of the changes made as well as line information that links with the manuscript.

MAJOR

The title is misleading and should be revised to something more accurate, such as “Effects of fasting and ex vivo hypoxia on murine skeletal muscle contractile function.” The authors don’t have to use this exact wording, but “ischemia” should not be used, “hypoxia” should be used, and the function should be specified.

-We agree with this concern and have revised the title and main body text to reflect these changes:

New title: Lines 1-2 “Effects of fasting on isolated murine skeletal muscle contractile function during acute hypoxia”. 

Lines: 26-27 Example of use in revised text: “Whether these constraints affect overall functional capacity or the timing of muscle energetic failure during acute hypoxia is not known”.

Ischemia refers to low blood flow. The muscles are studied ex vivo without any flow, whether they are oxygenated or not. Low oxygen is not the only consequence of ischemia. There is no convincing evidence that this is a good model for ischemia. The repeated use of the word “ischemia” or “ischemic” to describe the experiment should be eliminated. The experiment is studying hypoxia, not ischemia, and the text in the entire manuscript should be revised accordingly. Eliminating or at least deemphasizing the assertion that the experimental approach is an ischemia model would be helpful. If the authors are determined to comment on how this model has relevance to ischemia, they need to provide specific and direct evidence to support this assertion, and to also directly acknowledge the limitations of this experimental approach as an ischemia model.

-We have revised the text to avoid any misinterpretation of the data by changing the word ‘ischemia’ to ‘hypoxia’ where it refers directly to the model or its interpretation. Example of use in the text: 

Example Lines78-80 “We hypothesized that fasting, and associated reductions in stored muscle glycogen, would significantly shorten the amount of time that the muscles could remain functional during acute hypoxia”. 

The abstract refers to “conditions of reduced carbohydrate supply” before using a more informative description of “fasting.” The abstract should identify the duration of fasting. The text throughout should also not suggest that all fasting does is reduce carbohydrate supply or glycogen levels. It is OK to indicate that this might be an important consequence of fasting for the effects on contraction function, but there should be a more accurate description of what fasting represents and recognition that reducing glycogen is not everything it does.

-Excellent point. We have revised the abstract to reflect these changes. 

Lines: 27-32 “We interrogated skeletal muscle contractile properties in two anatomically distinct hindlimb muscles that have well characterized differences in energetic efficiency (locomotory- extensor digitorum longus (EDL) and postural- soleus muscles) following a 24-hour fasting period that resulted in substantially reduced muscle carbohydrate supply”. 

Additional lines: 79-81 “We hypothesized that fasting, and associated reductions in stored muscle glycogen, would significantly shorten the amount of time that the muscles could remain functional during acute hypoxia”.

A major point made by the authors is that glycogen concentration is much higher in the soleus than the EDL. This result has not been observed in earlier research. Glycogen of soleus was not much greater for mouse soleus compared to EDL (Jorgensen J Biol Chem. 2004. 279(2):1070-9; Bonen J Appl Physiol 1994. 76(4):1753-8). The authors should address what might account for the discrepant results and provide evidence that their results are consistent with results of a number of earlier studies.

-We thank the reviewer for this suggestion, and agree that additional support from the literature will strengthen the interpretation of our data. We have added several additional references to the manuscript:

• Ryder JW, Kawano Y, Galuska D, Fahlman R, Wallberg-Henriksson H, Charron MJ, et al. Postexercise glucose uptake and glycogen synthesis in skeletal muscle from GLUT4-deficient mice. FASEB J. 1999;13(15):2246–56. 

• Sandström ME, Abbate F, Andersson DC, Zhang SJ, Westerblad H, Katz A. Insulin-independent glycogen supercompensation in isolated mouse skeletal muscle: Role of phosphorylase inactivation. Pflugers Arch Eur J Physiol. 2004;448(5):533–8. 

• Jørgensen SB, Viollet B, Andreelli F, Frøsig C, Birk JB, Schjerling P, et al. Knockout of the α2 but Not α1, 5′-AMP-activated Protein Kinase Isoform Abolishes 5-Aminoimidazole-4-carboxamide-1-β-4-ribofuranoside- but Not Contraction-induced Glucose Uptake in Skeletal Muscle. J Biol Chem. 2004;279(2):1070–9. 

• Hunter RW, Treebak JT, Wojtaszewski JFP, Sakamoto K. Molecular mechanism by which AMP-activated protein kinase activation promotes glycogen accumulation in muscle. Diabetes. 2011;60(3):766–74. 

• Azpiazu I, Manchester J, Skurat A V., Roach PJ, Lawrence JC. Control of glycogen synthesis is shared between glucose transport and glycogen synthase in skeletal muscle fibers. Am J Physiol - Endocrinol Metab. 2000;278(2 41-2):234–43. 

• Helander I, Westerblad H, Katz A. Effects of glucose on contractile function, [Ca 2+] i, and glycogen in isolated mouse skeletal muscle. Am J Physiol - Cell Physiol. 2002;282(6 51-6):1306–12. 

• Bonen A, Mcdermott J, Tan M. Glycogenesis and Glyconeogenesis in Skeletal: Effects of pH and Hormones. Am J Physiol - Endocrinol Metab. 1990;258(4):693–700. 

• Bonen A, Homonko DA. Effects of exercise and glycogen depletion on glyconeogenesis in muscle. J Appl Physiol. 1994;76(4):1753–8.

We have also modified the discussion to include the following paragraph: 

Lines: 436-457 “Overnight fasting in rodents results in more dramatic metabolic effects than human overnight fasting, but induces experimentally reproduceable reductions in systemic carbohydrate stores that are similar to more extreme physiological conditions such as hyperinsulinemia, hypoglycemia, or post exercise recovery(11,14,29,39). Fasting was used in this study because it is independent of the confounding effects of exercise or contraction induced fatigue(30). There is a range of reported glycogen values available in the literature for mouse skeletal muscle and liver tissues which are likely influenced by assay method, normalization factor, as well as genetic background and physiological state of the animals(40–45). The fed state values observed for mouse soleus and EDL muscles in this study fall well within the range of normal variability observed in the literature(40–47). Additionally, the magnitude of tissue glycogen reduction observed with 24-hour fasting were concomitant with other available data(29)”. 

The muscle glycogen concentrations are higher than usually reported for mouse EDL and soleus. The value in Table 1 for fed soleus (61.9 nmol/mg) is very high compared to the literature. There should be citations of literature values for glycogen and an explanation for the high values in this study compared to the literature.

-Addressed in response to the previous comment above. We agree that the values may be interpreted as high compared to some studies, but there are other studies available that report even higher values. Ultimately the absolute values reported in the literature are likely very sensitive to physiological (e.g. insulin/glucagon stimulation, stress responses, diurnal cycle, etc.) and technical variables (e.g. assay method, normalization factor, etc.). To address this concern, we have added the references indicated above to highlight the range of values observed in other studies, and modified the language in our discussion to reflect the variability of reported absolute values:

Lines: 441-447 “There is a range of reported glycogen values available in the literature for mouse skeletal muscle and liver tissues which are likely influenced by assay method, normalization factor, as well as genetic background and physiological state of the animals(40–45). The fed state values observed for mouse soleus and EDL muscles in this study fall well within the range of normal variability observed in the literature(40–47). Additionally, the magnitude of tissue glycogen reduction observed with 24-hour fasting were concomitant with other available data(29)”.

The light/dark cycle times should be stated, and the times when fasting began and when muscles were sampled should be stated.

-Excellent suggestion. We apologize for this oversight. We have revised the methods section to reflect these changes: 

Lines: 141-146 “Mice were housed in a temperature-controlled facility on a 12-hour light dark cycle with free access to food and water prior to fasting. Mice were fasted for 24 hours to achieve a reduction of skeletal muscle glycogen of ~50% (compared to the fed state). The 24-hour fasting period began at the beginning of a light cycle and was terminated at the end of the subsequent dark cycle. Mice had free access to water during fasting. Muscles were isolated for experiments immediately following the end of the fasting period.”. 

The Methods section (lines 148-150) on fasting refers to a pilot study and cites a study (ref 23) that is not from this group of authors. It is confusing to know if the authors performed a pilot study or not, and why they cited this study.

- We apologize for the confusing way that this was presented. We modified the methods section to clarify the procedure: 

Lines: 141-146 “Mice were housed in a temperature-controlled facility on a 12-hour light dark cycle with free access to food and water prior to fasting. Mice were fasted for 24 hours to achieve a reduction of skeletal muscle glycogen of ~50% (compared to the fed state). The 24-hour fasting period began at the beginning of a light cycle and was terminated at the end of the subsequent dark cycle. Mice had free access to water during fasting. Muscles were isolated for experiments immediately following the end of the fasting period”.

In the statistics section (lines 242-243), it stated that both SEM and SD are used with the data. Either one or the other should be used. SD is more informative.

-We apologize for the confusing way that this was presented in the methods. We have modified the methods section to reflect only use only sample standard deviation:

Line: 238 “Data are represented by mean ± sample standard deviation (SD)”.

The Discussion should acknowledge important limitations of the study. One would be that only one timepoint was studied for metabolite concentrations. Measurements at several timepoints would make the study more informative.

-This is an excellent point. We have added additional discussion of the limitations of our study design: 

Lines: 490-502 “There are several important limitations to this study. First, the carbohydrate reduction associated with fasting is not complete, leaving approximately half of the fed state muscle glycogen available during experimental hypoxia. Though this is independent of confounding effects associated with other methods of glycogen depletion(30), there are other effects of fasting that may confound observed outcomes(29). Second, though we were able to assess muscle contractile function in time series, we were only able to assess changes in muscle metabolite levels (e.g. adenine nucleotides, glycogen, etc.) at baseline and after the 180-minute protocols. Additional experimentation is necessary to fully characterize the time-dependent changes in key metabolites during hypoxia. Finally, our experimental system utilizing isolated muscle is highly controlled for the effects of hypoxia but lacks the biological variability and complexity of ischemia. We hope that the observations in this study can be used to inform development of hypotheses that can be further tested using in vivo preclinical models of ischemia”. 

The final sentence of the Introduction is that the “This information …ischemia models.” The Conclusion states that the results are valuable for therapeutic intervention (lines 479 and 488). It is unclear why this information will be valuable for either these ischemia models or for therapeutic interventions. It should be directly stated why this information will be valuable.

-We apologize for any apparent over-interpretation of the data. We have modified the introduction and discussion to clarify our intent and interpretation: 

Lines: 86-90 “Our data provide a novel characterization of hypoxic muscle mechanical/energetic failure and paint a detailed picture of the timing of these impairments. This information can be used in conjunction with existing in vivo rodent hindlimb ischemia/reperfusion studies(5,16,17,19) to generate new hypotheses regarding optimal timing of reperfusion or administration of precision therapeutics”. 

Additional lines: 40-42 “Fasting resulted in greater passive tension development in both muscle types, which may have implications for the design of pre-clinical studies involving optimal timing of reperfusion or administration of precision therapeutics”. 

Figure 4 should include text on the figure itself to indicate which A-F panels are from the O2 treatment and which are from N2 treatment.

-We apologize for this oversight. We have reorganized the panels in this figure to include the basal, O2, and N2 conditions within each graph for each nucleotide. The feeding conditions and bath conditions are both indicated within each graph. Figure 4A-H. Additionally, we have added cross-hatch patterns to the fasted group bars to help differentiate between the two levels. Finally, we have included individual data points in order to enhance visualization of the data variability. 

In the Introduction (line 76), it stated that the experiment was intended to determine the “exact temporal nature…”, but only one time point (3 hours) was studied, so this study doesn’t determine the “exact temporal nature” of the results.

-We thank the reviewer for this insight and appreciate the concern. Though we were only able to measure metabolite levels at the terminal endpoint of the experiments, we interpret the force production and work measurements to be informative of the overall energetic state of the muscle during the time course of the experiments. We have modified the introduction to clarify our intent and interpretation of the limitations of the study: 

Lines: 76-79 “This led us to examine the <3-hour time domain in this study to better define the timing of muscle functional impairments and associated terminal metabolite changes that occur during acute hypoxia”.

---

## [Decision Letter · Decision Letter 1]

10 Feb 2020

PONE-D-19-31637R1

Effects of fasting on isolated murine skeletal muscle contractile function during acute hypoxia.

PLOS ONE

Dear Dr. McClung,

Thank you for submitting your manuscript to PLOS ONE. After careful consideration, we feel that it has merit but does not fully meet PLOS ONE’s publication criteria as it currently stands. Therefore, we invite you to submit a revised version of the manuscript that addresses the points raised during the review process.

Although some improvements have been made to the manuscript the measurement of glycogen is still a major concern. In order for this manuscript to be suitable for publication all of the reviewer’s comments must be specifically addressed.    

We would appreciate receiving your revised manuscript by Mar 26 2020 11:59PM. To enhance the reproducibility of your results, we recommend that if applicable you deposit your laboratory protocols in protocols.io, where a protocol can be assigned its own identifier (DOI) such that it can be cited independently in the future. For instructions see: http://journals.plos.org/plosone/s/submission-guidelines#loc-laboratory-protocols

We look forward to receiving your revised manuscript.

Kind regards,

Cameron J. Mitchell, PhD

Academic Editor

PLOS ONE

Reviewers' comments:

Reviewer's Responses to Questions

**Comments to the Author**

1. If the authors have adequately addressed your comments raised in a previous round of review and you feel that this manuscript is now acceptable for publication, you may indicate that here to bypass the “Comments to the Author” section, enter your conflict of interest statement in the “Confidential to Editor” section, and submit your "Accept" recommendation.

Reviewer #1: (No Response)

Reviewer #2: (No Response)

2. Is the manuscript technically sound, and do the data support the conclusions?

Reviewer #1: (No Response)

Reviewer #2: Partly

3. Has the statistical analysis been performed appropriately and rigorously? 

Reviewer #1: (No Response)

Reviewer #2: Yes

4. Have the authors made all data underlying the findings in their manuscript fully available?

Reviewer #1: (No Response)

Reviewer #2: (No Response)

5. Is the manuscript presented in an intelligible fashion and written in standard English?

Reviewer #1: (No Response)

Reviewer #2: No

6. Review Comments to the Author

Reviewer #1: (No Response)

Reviewer #2: The authors made a number of revisions that have improved the manuscript. However, they have not adequately addressed issues related to the questions about the glycogen results. The current text is not forthcoming in acknowledging or explaining the discrepant glycogen values in this manuscript compared to the literature. The specific concerns are described below.

The authors’ response did not address the fact that their observation of much higher glycogen in the soleus than the EDL, and this result has not been observed in earlier research. Glycogen of soleus was not much greater for mouse soleus compared to EDL (Jorgensen J Biol Chem. 2004. 279(2):1070-9; Bonen J Appl Physiol 1994. 76(4):1753-8). The authors should address what might account for the discrepant results and provide evidence that their results are consistent with results of a number of earlier studies. The authors cited the 2 publications above (which found soleus values were slightly, but not significantly lower for soleus versus EDL) and 6 additional publications, none of which reported glycogen for both soleus and EDL in mice. The authors did not acknowledge that their results of 80% greater glycogen in soleus compared to EDL are at odds with the published literature, and they offered no data that supported their discrepant results. There needs to be a direct explanation in the manuscript related to the unusual findings in this study.

The value for soleus glycogen in Table 1 (61.9 nmol/mg) is much greater than previously published values. The authors state that other studies have reported even higher values, but they don’t point out which studies reported higher values. In the revised manuscript, they state the observed values “in this study fall well within the range of normal availability observed in the literature”, and they cite 8 publications (#40-47). However, several of these publications do not report mouse soleus or EDL glycogen values, and the studies that do report glycogen in these muscle do not report values higher than 61.9 nmol/mg. Some cited studies only reported gastrocnemius glycogen, or single fiber glycogen, or not glycogen concentrations (only glycogenesis rates). Citations that do not include mouse soleus and or EDL glycogen concentrations should not be cited to support the statement in the manuscript. The text needs to be revised to be accurate. The text refers to values being influenced by “assay method, normalization factor, as well as genetic and physiological state of the animals.” This statement is true, but it doesn’t explain the much greater value in this study. If there is any specific evidence that the genetic or physiological state of the mice, or the particular assay used in this study led to much increased soleus glycogen values, it should be specifically identified rather than this general statement without any direct support of its validity.

7. PLOS authors have the option to publish the peer review history of their article (what does this mean?). If published, this will include your full peer review and any attached files.

Reviewer #1: No

Reviewer #2: No

---

## [Author Response · Author response to Decision Letter 1]

20 Mar 2020

We sincerely apologize for failing to appropriately address the concerns regarding the reported muscle glycogen values. The references included in the last resubmission were chosen to highlight a range of potential factors that reflect variation in reported values for mouse skeletal muscle glycogen, but clearly, we missed the mark. We appreciate the reviewer’s concerns and thank them for their ongoing efforts in aiding our improvement of the quality of this manuscript. Below, we attempt to address all the key concerns that we interpret from the provided comments. 

1. “The authors should address what might account for the discrepant results”

We have added additional text to the discussion. Lines 448-458. “Though we cannot directly account for specific confounders that explain the discrepancy in this study, there are two likely candidates that should be considered for further investigation: 1.) Muscle glycogen levels have been shown to vary with season and diurnal cycle in mice(44) and rats(45,46), with peaks in the dark-light cycle transition period (the time at which animals were sacrificed in this study). 2.) Soleus muscles have been shown to be more sensitive to insulin stimulated glucose uptake and glycogen synthesis in both mice(38) and rats(47), and insulin stimulated glucoregulatory responses have been shown to differ among inbred mouse strains(48). Taken together, the described findings support the possibility that stored muscle glycogen values may have been influenced by seasonal, circadian, or hormonal variation intrinsic to the genetic background of the mice used in this study”.

2. “provide evidence that their results are consistent with results of a number of earlier studies”

Our observed glycogen values differed from three prior studies that assessed mouse EDL and soleus muscle glycogen content using similar assay methods (cited in the paper and listed below; these were the best matched studies we could find in terms of methodology and including both EDL and soleus muscles). We have altered the manuscript to clearly acknowledge that the results differ from the cited studies and added discussion of potential confounding variables (see response described above). We have found it difficult to directly account for the discrepancies. However, there are a few methodological and environmental differences between this study, and the cited references (particularly contribution of diurnal variation, seasonal variation, and parental genetic background variation as discussed). We have been fully transparent regarding our methods/data analysis and feel that the methodological description in this manuscript is thorough and comparable to all of the published studies that we have reviewed. Additionally, all supporting raw data will be available to the scientific community via open science framework if the manuscript is accepted for publication (in accordance with Plos One’s publishing requirements). We value the importance of ensuring that the data are of high quality and reproducibility. However, we also appreciate that biological data is variable and subject to substantial influence from a multitude of confounding factors. If there are any additional specific methodological concerns, we will enthusiastically address them. We don’t want to risk sharing inaccurate data, but we also want to avoid denying the scientific community access to data solely on the basis that it differs from previous findings.

Some additional technical considerations: For brevity, we reported overall sample size for the groups by using the smallest group size (as this has the largest effect on the ANOVA power). The actual sample sizes for the fed state basal EDL and soleus is N=17 and N=10 respectively. This occurred because of additional myography (twitch protocols) and biochemical assays (UPLC amino acid profiles) that ultimately were not included in the manuscript in order to keep the supporting evidence concise. The soleus glycogen data does have two outliers (87.2 and 96.2 nmol/mg) that skew the mean toward a higher value, however, we did not have any reasonable evidence that the values should be excluded from the data set (61.9±17.5nmol/mg with outliers; 54.4±8.6 nmol/mg without outliers; Mean±SD). Even if the outliers were excluded, the soleus glycogen would have been higher than the EDL values (34.4±8.0 nmol/mg), and both were higher than the values reported in the cited references. 

We do not have the original lysates used to determine the glycogen values from the original study and re-performing all of the assays would necessitate re-performing the entire study. However, in the interest of ensuring that our assay methods were sound, we did attempt to assess the reproducibility of the assay. For this quality control (QC) assay, we performed additional glycogen measurements in muscles isolated from BALB/c mice. However, the mice that we had available were younger and of the opposite sex than those used in the study, therefore it is difficult to rule out contributions of biological variability from the reproduced assays. We were able to match the circadian timing for sacrifice, but not seasonal timing (Summer for study vs. winter for QC). However, we provide the data for comparison below. Though the higher glycogen values in the soleus observed in the study were not reproduced, we interpret the consistent values for the tibialis anterior muscles to indicate that the assay is sound and reproducible, and that the previously observed discrepancy was likely derived from biological variation rather than technical variation. Some of the difference in values between study and QC values may also be attributable to large differences in muscle size between the two groups. In both cases the samples were run on the same plate, in triplicate, with R2 of the standard curve ≥.99.

Muscle Sample size (N) Mean Glycogen (nmol/mg tissue wet weight) Standard deviation

Tibialis Anterior (study) 18 38.7 6.7

Tibialis Anterior (QC Assay) 6 32.3 2.6

EDL (study) 17 34.4 8.0

EDL (QC) 6 23.1 2.4

Soleus (Study) 10 61.9 17.5

Soleus (QC) 6 27.3 6.7

3. “The authors did not acknowledge that their results of 80% greater glycogen in soleus compared to EDL are at odds with the published literature, and they offered no data that supported their discrepant results”

We have added additional discussion to the manuscript to fully acknowledge the discrepancy and to try to explain potential sources of variability in the data. Lines 436-455 “Overnight fasting in rodents results in more dramatic metabolic effects than human overnight fasting, but induces experimentally reproduceable reductions in systemic carbohydrate stores that are similar to more extreme physiological conditions such as hyperinsulinemia, hypoglycemia, or post exercise recovery(1–4). Fasting was used in this study because it is independent of the confounding effects of exercise or contraction induced fatigue(5). Notably, the glycogen values observed in this study differ from several previous reports in that fed state control values for both muscles are relatively high, and that the soleus glycogen levels are substantially higher than the EDL (values between muscles did not differ in the previous reports)(6–8). 

Muscle glycogen concentration is a physiologically dynamic parameter that is influenced by experimental conditions such as assay method and normalization factor, as well as biological conditions such as parental genetic background and metabolic state (9–11). Though we cannot directly account for specific confounders that explain the discrepancy in this study, there are two likely candidates that should be considered for further investigation: 1.) Muscle glycogen levels have been shown to vary with season and diurnal cycle in mice(12) and rats(13,14), with peaks in the dark-light cycle transition period (the time at which animals were sacrificed in this study). 2.) Soleus muscles have been shown to be more sensitive to insulin stimulated glucose uptake and glycogen synthesis in both mice(6) and rats(15), and insulin stimulated glucoregulatory responses have been shown to differ among inbred mouse strains(16). Taken together, the described findings support the possibility that stored muscle glycogen values may have been influenced by seasonal, circadian, or hormonal variation intrinsic the genetic background of the mice used in this study”.

Additional Lines 486-498 “In this study, we observed that Soleus muscles stored more glycogen at baseline, had greater specific force/work capacities, and produced absolute force for a longer period during hypoxia compared to EDL muscles. The observations regarding greater glycogen content in the soleus muscle compared to EDL muscles are not consistent with previous reports(7,8,17), but the observations of improved mechanical function during hypoxia in soleus compared to EDL muscles have been previously reported using small muscles isolated from rats(18). Though the absolute differences in glycogen concentrations between groups were larger in the soleus compared to the EDL, the response coefficient (RGlyc) which facilitates interpretation of group differences relative to their baseline concentration, indicated that the patterns of utilization were not different between the two types of muscles. We interpret these findings to mean that the greater basal glycogen concentration observed in the soleus muscles was likely not the primary factor underlying it’s enhanced ischemic mechanical performance“. 

4. “The authors state that other studies have reported even higher values, but they don’t point out which studies reported higher values”

a. We apologize for any confusion regarding our interpretation of the additional references provided in the previous submission. Our intent was to show that the measured values fell within a reasonable range of values relative to previous reports (i.e. that the values were physically possible). We have been careful to ensure that any specific references to published data are clearly linked to relevant text (see above paragraphs and additional references). Additionally, we have removed any references that do not directly reference the EDL and/or Soleus muscles. 

5. “Citations that do not include mouse soleus and or EDL glycogen concentrations should not be cited to support the statement in the manuscript.”

a. We have removed two of the cited references that did not specifically assess glycogen content in mouse EDL and/or soleus muscles.

6. “If there is any specific evidence that the genetic or physiological state of the mice, or the particular assay used in this study led to much increased soleus glycogen values, it should be specifically identified rather than this general statement without any direct support of its validity.”

a. As discussed, the discrepancies in the measured values are difficult to account for directly. However, there are a few candidate sources of variation that have been more thoroughly discussed in the manuscript in an attempt to account for the differences. These include: references describing seasonal and diurnal variation in muscle glycogen content in mice and rats. Diurnal variation in muscle glycogen typically peaks at the end of the dark cycle/beginning of the light cycle, when the animals in this study were sacrificed. Thus, we think it reasonable to suggest that this may be a confounding variable here. Additionally, soleus muscles in mice and rats have been shown to be more sensitive to insulin stimulated glucose uptake.

---

## [Decision Letter · Decision Letter 2]

2 Apr 2020

PONE-D-19-31637R2

Effects of fasting on isolated murine skeletal muscle contractile function during acute hypoxia.

PLOS ONE

Dear Dr. McClung,

Thank you for submitting your manuscript to PLOS ONE. After careful consideration, we feel that it has merit but does not fully meet PLOS ONE’s publication criteria as it currently stands. Therefore, we invite you to submit a revised version of the manuscript that addresses the points raised during the review process.

We would appreciate receiving your revised manuscript by May 17 2020 11:59PM. To enhance the reproducibility of your results, we recommend that if applicable you deposit your laboratory protocols in protocols.io, where a protocol can be assigned its own identifier (DOI) such that it can be cited independently in the future. For instructions see: http://journals.plos.org/plosone/s/submission-guidelines#loc-laboratory-protocols

We look forward to receiving your revised manuscript.

Kind regards,

Cameron J. Mitchell, PhD

Academic Editor

PLOS ONE

Additional Editor Comments (if provided):

Before the manuscript can be published please address the minor comments raised by reviewer 2.

Reviewers' comments:

Reviewer's Responses to Questions

**Comments to the Author**

1. If the authors have adequately addressed your comments raised in a previous round of review and you feel that this manuscript is now acceptable for publication, you may indicate that here to bypass the “Comments to the Author” section, enter your conflict of interest statement in the “Confidential to Editor” section, and submit your "Accept" recommendation.

Reviewer #2: (No Response)

2. Is the manuscript technically sound, and do the data support the conclusions?

Reviewer #2: Yes

3. Has the statistical analysis been performed appropriately and rigorously? 

Reviewer #2: I Don't Know

4. Have the authors made all data underlying the findings in their manuscript fully available?

Reviewer #2: Yes

5. Is the manuscript presented in an intelligible fashion and written in standard English?

Reviewer #2: Yes

6. Review Comments to the Author

Reviewer #2: Because circadian and seasonal effects are described as important for glycogen values, please specify for each experiment (if is the same for every experiment, that can be noted and it can be stated only once):

1) The times in the day when lights were turned on and turned off.

2) The times in the day when samples were collected.

3) The month(s) when experiments were performed,

7. PLOS authors have the option to publish the peer review history of their article (what does this mean?). If published, this will include your full peer review and any attached files.

Reviewer #2: No

---

## [Author Response · Author response to Decision Letter 2]

3 Apr 2020

Reviewer #2: Because circadian and seasonal effects are described as important for glycogen values, please specify for each experiment (if is the same for every experiment, that can be noted and it can be stated only once):

1) The times in the day when lights were turned on and turned off.

2) The times in the day when samples were collected.

3) The month(s) when experiments were performed

We thank the reviewer for their continued time and effort. We appreciate the constructive nature of the feedback and feel that the manuscript has been substantially improved by our exchanges. To address the outlined concerns, we have added additional details to the methods section:

Lines 141-149 “Mice were housed in a temperature-controlled facility on a 12-hour light-dark cycle with free access to food and water prior to fasting (dark cycle: began at 1900 hours, ended at 0700 hours). Mice were fasted for 24 hours to achieve a reduction of skeletal muscle glycogen of ~50%, compared to the fed state. The 24-hour fasting period started at the beginning of a light cycle (0700 hours) and was terminated at the end of the subsequent dark cycle (0700 hours). Mice had free access to water during fasting. All muscles (including control and fasted groups) were isolated for experiments immediately following the end of the dark cycle, between 0700 and 0800 hours. All experiments were performed in the summer season, between the months of May and August”.

---

## [Editor Report · Decision Letter 3]

6 Apr 2020

Effects of fasting on isolated murine skeletal muscle contractile function during acute hypoxia.

PONE-D-19-31637R3

Dear Dr. McClung,

We are pleased to inform you that your manuscript has been judged scientifically suitable for publication and will be formally accepted for publication once it complies with all outstanding technical requirements.

With kind regards,

Cameron J. Mitchell, PhD

Academic Editor

PLOS ONE
---

## [Editor Report · Acceptance letter]

10 Apr 2020

PONE-D-19-31637R3 

Effects of fasting on isolated murine skeletal muscle contractile function during acute hypoxia. 

Dear Dr. McClung:

I am pleased to inform you that your manuscript has been deemed suitable for publication in PLOS ONE. Congratulations! Your manuscript is now with our production department. 

With kind regards,

on behalf of

Dr. Cameron J. Mitchell 

Academic Editor

PLOS ONE